# Investigating Color-Blind User-Interface Accessibility via Simulated Interfaces †

Amaan Jamil and Gyorgy Denes *

The Perse School, Cambridge CB2 8QF, UK
* Correspondence: gdenes@perse.co.uk
† This paper is an extended version of our paper published in Computer Graphics & Visual Computing 2023, Wales, UK, 14–15 September 2023.

**Abstract:** Over 300 million people who live with color vision deficiency (CVD) have a decreased ability to distinguish between colors, limiting their ability to interact with websites and software packages. User-interface designers have taken various approaches to tackle the issue, with most offering a high-contrast mode. The Web Content Accessibility Guidelines (WCAG) outline some best practices for maintaining accessibility that have been adopted and recommended by several governments; however, it is currently uncertain how this impacts perceived user functionality and if this could result in a reduced aesthetic look. In the absence of subjective data, we aim to investigate how a CVD observer might rate the functionality and aesthetics of existing UIs. However, the design of a comparative study of CVD vs. non-CVD populations is inherently hard; therefore, we build on the successful field of physiologically based CVD models and propose a novel simulation-based experimental protocol, where non-CVD observers rate the relative aesthetics and functionality of screenshots of 20 popular websites as seen in full color vs. with simulated CVD. Our results show that relative aesthetics and functionality correlate positively and that an operating-system-wide high-contrast mode can reduce both aesthetics and functionality. While our results are only valid in the context of simulated CVD screenshots, the approach has the benefit of being easily deployable, and can help to spot a number of common pitfalls in production. Finally, we propose a AAA–A classification of the interfaces we analyzed.

**Keywords:** user interface; color vision; color blindness; CVD; accessibility

## 1. Introduction

Color vision deficiency (CVD, color blindness) is the failure or decreased ability to distinguish between colors under normal illumination. There are over 300 million people with CVD, including approximately 1 in 12 men (8%) and 1 in 250 women (0.5%) [1–3]. CVD is an X-linked genetic disorder impacting both eyes, with varying degrees of prevalence in different populations [4]. It affects an individual's ability to perform tasks in both personal and professional settings [5].

Color is an important asset in user-interface (UI) design [6], and while the exact impact of color is known to vary between demographics [7], applications still often rely on established conventions, such as green and red indicating 'yes' and 'no', respectively. Objects of the same color satisfy the Gestalt principle of similarity, whereas different colors can help an object stand out or mark figure–ground articulation [8]. With the ever-increasing color gamut of novel displays [9], new domains are opening up in the use of color; however, some of these domains simply cannot be seen by someone with CVD.

Accessibility is the concept of making UIs equally usable by all types of users, enabling user interactions without barriers. UI designers have the option to support accessibility for CVD users pre-publication or post-publication [10], with pre-publication normally resorting to a limited and fixed color palette [2], and post-production relying on automatic

recoloring [11] also known as *daltonization* [12]. A hybrid, low-effort approach is to provide support for the operating system's high-contrast mode. The Web Content Accessibility Guidelines (WCAG) [6] outline some best practices for accessibility; however, these often only target core functionality. Another consideration for UI is perceived aesthetics. Many designers see aesthetics as inversely proportional to functionality [13], while other evidence points towards a positive correlation between functionality and aesthetics [14–16]. In the domain of CVD, a high-contrast theme is rarely a top-priority feature, which could imply that people with CVD have a substantially reduced aesthetic experience. However, there are insufficient data to understand CVD users' perceived functionality and aesthetics of UIs.

A comparative study of UI functionality and aesthetics is inherently challenging. Individuals with CVD cannot judge if a reduced-dimensionality UI they see is equally usable or aesthetic compared to a UI they could never see. Therefore, we instead built on the successful field of physiologically based CVD simulations and asked 19 non-CVD participants to compare reference UIs to how they might appear to a CVD observer for 20 popular UIs (1449 data points in total). Specifically, we measured mean-opinion scores for functionality and the probability of maintained aesthetics. One clear limitation of our study is non-CVD users' potential bias towards familiar full-color references; therefore, our results cannot be used to conclude whether any of the UIs are fully accessible. However, comparing scores across different applications and accessibility techniques, we can investigate common pitfalls of UI design. Our main contributions can be summarized as follows:

- Collection of subjective user data from non-CVD observers on the loss of *functionality* and *aesthetics* when seen through CVD simulation;
- Analysis of results suggesting a positive correlation between functionality and aesthetics scores, and evidence that OS-enabled high-contrast mode might be detrimental;
- Publication of the dataset and participant responses.

The rest of the paper is structured as follows: in Section 2, we discuss the relevant background literature; in Section 3, we describe the proposed experimental methodology, before discussing our findings in Sections 4 and 5.

## 2. Background

### 2.1. CVD

Human color vision is widely considered trichromatic due to the three types of retinal cone cells ($L$, $M$, and $S$) that respond differently to wavelengths of light (Figure 1) [17]. According to opponent color theory [18], higher-order visual functions rely on a differential response of cones, with luminance ($L + M$) being more prominent than chrominance channels ($L - M$, $L + M - S$). CVD can occur as a result of cone cells being absent, not working, or having an abnormal response. Severe CVD occurs when two or three types of cone cells are absent (monochromacy, achromatopsia). Less severe CVD occurs when one type of cone is absent ($L$: protanopia, $M$: deuteranopia, $S$: tritanopia), or all three are present but one cell has an abnormal response ($L$: protanomaly, $M$: deuteranomaly, $S$: tritanomaly) [19]. Red–green CVD, specifically protanomaly and deuteranomaly, are the most frequent [20], which can be intuitively explained by the spectral similarity of the $M$ and $L$ cones. This affects approximately 8% and 0.1–0.3% of the male and female populations, respectively [1].

Color appearance is context-dependent, highly subjective [21], and, especially as color plays such a key role in visual perception, most find it challenging to imagine how another individual (especially with CVD) might perceive the world around them. However, the physiological mechanisms of early vision are relatively well understood, and existing models have been shown to be capable of faithfully simulating what a CVD observer might see [22]. We build on these models in our experimental methodology.

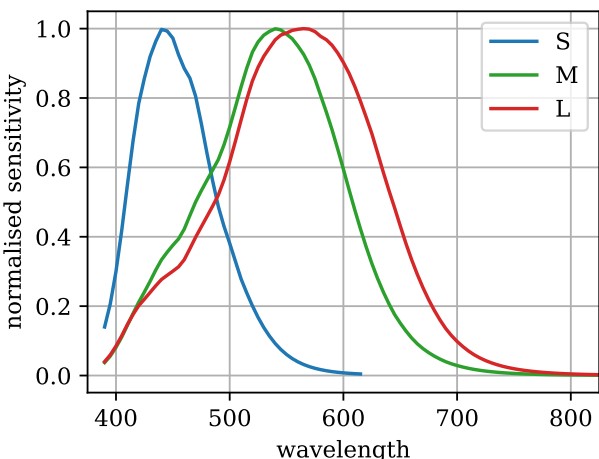

**Figure 1.** Normalized spectral sensitivity of the three types of retinal cone cells as a function of light wavelength. Note the similarity of the *M* and *L* cones, where minor individual differences can result in an unresolvable overlap, causing observers to have a reduced ability to differentiate between lights in the red–green region. This is plotted based on Stockman and Sharpe [23].

*2.2. WCAG*

Accessibility is the concept of making UIs equally usable by all types of users, enabling all user interactions without barriers. In our work, we consider *functionality* to be synonymous with *usability*; therefore, accessibility also aims to maintain functionality. In the past few decades, accessibility has become a legal requirement in several countries [24], with most government agencies recommending the adoption of the World Wide Web Consortium's (W3C) Web Content Accessibility Guidelines (WCAG) [6,24]. In fact, WCAG is consulted by website and UI developers every day to enforce best practices to maintain a level playing field for all users, regardless of age, or physical or mental disabilities. WCAG has also been utilized in academic research to assess website accessibility [25–27].

WCAG 2.1 [6], the latest framework, specifies a set of *success criteria* in an **A**–**AAA** system, **A** being the least and **AAA** being the most restrictive. Minimum accessibility compliance is often set at **AA** [24]. CVD support is addressed across several criteria, and the following are given as examples:

- 1.4.1 Use of Color (**A**): encourages providing information conveyed via color through other visual means;
- 1.4.5 Images of Text (**AA**): encourages text being used to convey information rather than images of text;
- 1.4.11 Non-text Contrast (**AA**): visual presentation of UI components and graphical objects having a contrast ratio of at least 3:1;
- 1.4.6 Contrast (Enhanced) (**AAA**): visual presentation of text and images of text has a contrast ratio of at least 7:1.

It remains uncertain to what extent current UIs implement these recommendations. Also, while the use of *contrast ratio thresholds* gives designers an intuitive scheme, *contrast* is defined as a color difference perceived by trichromatic (non-CVD) observers, which is not trivially convertible to contrast for CVD users. Furthermore, WCAG contrast is typically calculated for background vs. foreground, which can fail to capture some impacts of CVD, such as reduced differences across UI components (Figure 2). We accept the importance and impact of WCAG, but we argue that a user study is more appropriate for CVD.

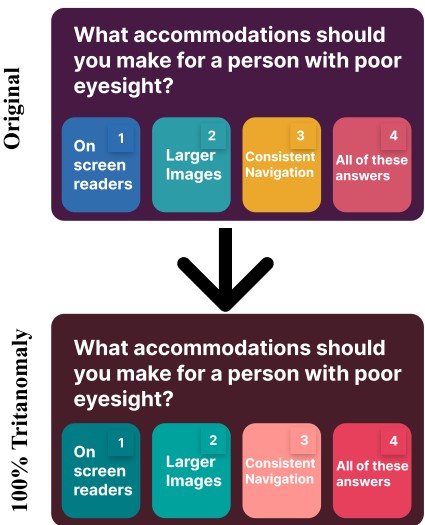

**Figure 2.** Illustration of a popular quiz platform's UI processed with simulated tritanomaly [22]. While text vs. background contrast remains mostly unchanged, $\Delta E_{2000}$ color difference [28] between buttons 1 and 2 is reduced from 25.1 to 14.1. Similarly, $\Delta E_{2000}$ between buttons 3 and 4 is reduced from 43.0 to 18.3. The colors remain distinguishable, but the difference between them is less noticeable. WCAG analysis would flag an insufficient text vs. background contrast on button 3 (<3), but the current guidelines do not address the reduced color difference between buttons.

### 2.3. Reducing the Impact of CVD

To reduce the impact of CVD, WCAG encourages UI designers to convey information via alternate visual channels. This also agrees with recent research on the use of patterns [29] and textures [30] to improve accessibility.

Another widely adopted approach limits the color palette with an increased contrast between key colors [2,10], e.g., Windows users can select from multiple operating-system-wide *high-contrast* themes that accommodate different types of vision deficiencies. While almost all applications and websites respect the high-contrast theme of the operating system, the resulting UI often looks substantially different, with seemingly reduced information for a non-CVD observer (see Figure 3). In our study, we investigate whether this strategy is viable by measuring functionality and aesthetics with the operating system's high-contrast mode enabled and disabled.

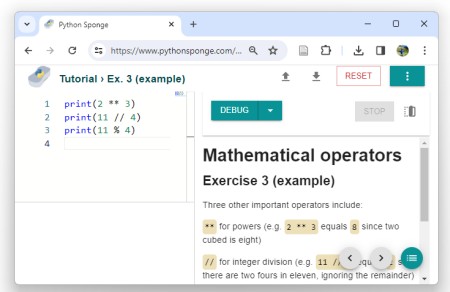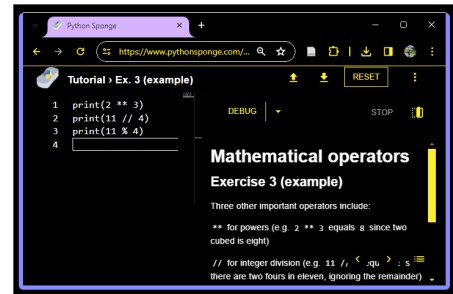

**Figure 3.** Illustration of a website automatically adapting to Windows high-contrast mode (*Night Sky*). The reduced color palette is designed to increase contrast between key UI components.

An alternative technique is *daltonization*, an umbrella term for image-based recoloring techniques that enhance displayed images for CVD observers [11,31,32]. Such algorithms aim to alter colors that are isochromatic to CVD observers but are perceived differently by a non-CVD observer. Daltonization could substantially improve CVD accessibility;

however, current algorithms are still limited by, for example, run-time cost, lack of temporal coherence, or the ability to accommodate different types and levels of CVD [33].

### 2.4. Aesthetics

While accessibility and overall functionality are key features of a good UI, designers, and human–computer interaction (HCI) researchers are increasingly discovering the impact of attractiveness and *beauty*. An attractive UI can help make a connection between a user and the application [34]. Interdisciplinary research has investigated the relevance of aesthetics, a classical philosophical field focusing on the "perception of the beautiful in nature and art" [35]. A particular focus has been the difference between classical and expressive aesthetics in HCI, with classical aesthetics referring to the traditional notions of an orderly and clear design (e.g., sizing), and expressive aesthetics referring to creativity and originality [36]. Mahlke and Thüring [37] argue that classical aesthetics is perceived more evenly, whereas expressive aesthetics varies with framing; as such, in this paper, we focus on classical aesthetics. For a more complete review of the literature, refer to Ahmed et al. [14].

The relationship between UI aesthetics and functionality has been investigated by a number of authors. Some argue that the two goals often conflict [13], whereas Kurosu and Kashimura [15] found a positive correlation between the *beauty* of ATM interfaces and their usability. Due to the subjective nature of aesthetics, the strength of the correlation coefficient has been shown to be culturally dependent [16]. To our knowledge, no previous publication has investigated aesthetics in the context of CVD. While our simulation-based experimental methodology is not robust enough to measure aesthetics for CVD people, it allows us to hypothesize that functionality and classical aesthetics correlate in CVD-simulated UIs, and that classical aesthetics could be negatively impacted by automatic invasive techniques, such as the high-contrast mode.

### 3. Materials and Methods

In our experiment, we aim to comparatively study the following:

- Functionality: how well could a CVD user use the interface? For example, compared to a non-CVD user, is usability reduced? Are key visual elements equally perceivable?
- Aesthetics: according to classical aesthetics, would the clarity and orderliness of the UI be retained when viewed by a CVD observer?

As discussed earlier, the experimental design is inherently non-trivial. Individuals with CVD cannot make a comparative judgment on whether the stimulus they perceive has the same functionality and aesthetics as a stimulus that they could never see. Conversely, an individual with no CVD does not have sufficient experience to rate how an individual with CVD might perceive a stimulus. A large-scale study with cross-population comparisons of subjective ratings might be considered; however, the subjective nature of aesthetics and the various experiences and consequent biases of the two populations might still prevent such studies from drawing reliable conclusions.

Therefore, we instead build on the successful field of physiologically based models of CVD simulation that allow non-CVD observers to inspect UIs as seen by CVD observers. Such models are based on a strong scientific understanding of early vision and have been verified by controlled user experiments [22]. An inherent limitation of this approach is non-CVD observers' bias towards the more familiar reference images; therefore, we do not consider the protocol to be suitable to judge whether the functionality and aesthetics would be fully retained for a CVD observer. However, we argue that relative scores across applications and CVD mitigation strategies still reveal crucial insights. Furthermore, a simulation-based pipeline has the benefit that it is easily deployable in a commercial environment.

For an overview of the methodology, see Figure 4. In the following subsections, we describe the stimuli, setup, task, participant sample set, and the statistical methods applied.

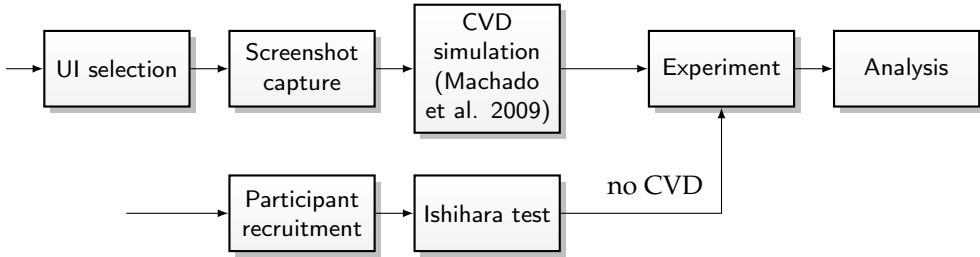

**Figure 4.** Overview of the experimental methods, with each step discussed in detail in the main text [22].

*3.1. Stimuli*

In our comparative study, we presented users with the original reference image as well as three *distorted* images that were identical in terms of content, showing what a CVD person might see. Figure 5 shows a high-level visualization of the interface; for a full-page screenshot, see Figure A2. The reference image was always presented in the top left; participants were made aware of this during the briefing, and there was a "Reference" label to remind them during the experiment. The distorted images were generated using CVD simulation from Machado et al. [22] as implemented in [38] assuming an sRGB display. We considered the three most frequent CVDs: protanomaly (top right), deuteranomaly (bottom left), and tritanomaly (bottom right). For the experiment, we set the simulation level conservatively to 100%.

For content, we sampled 20 software across different industries (social, business, entertainment, music, shopping, food and beverage, travel, education, productivity, and SaaS). Within each sector, we picked the application or website based on general popularity in the UK, according to informal surveys. This does not give full coverage of all available design techniques and CVD mitigations; however, it can serve as a representative sample of UIs that the average user is likely to encounter. The list of UIs includes Amazon, Booking.com, Brainly, Dropbox, Duolingo, Facebook, Google Calendar, Google Maps, Indeed, Instagram, LinkedIn, Photomath, Pocket, Slack, Spotify, Teams, Tiktok, Todoist, Uber Eats, and Zoom.

Participants viewed static screenshots. Specifically, in each software, we chose a common user journey that was relevant to the use of that software and recorded 3 screenshots. Some of these screenshots were resized with a maintained aspect ratio, and users could examine the original full-resolution content by hovering with the mouse.

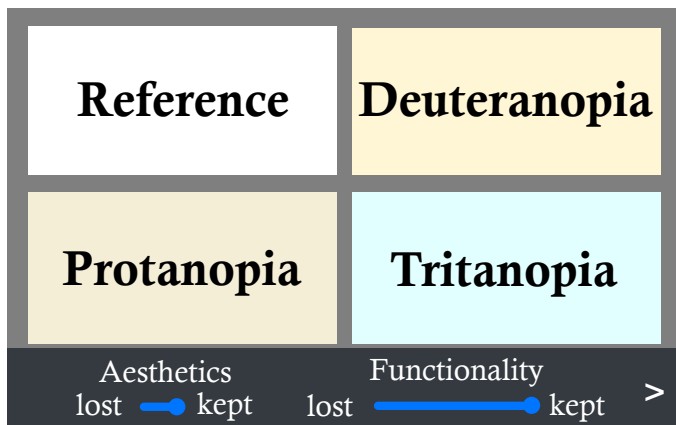

**Figure 5.** Visualization of the experiment setup with the reference screenshot and three types of simulated CVD (protanomaly, deuteranomaly, and tritanomaly) with otherwise identical content. The user controls the aesthetics (binary)and functionality scores (1–5) at the bottom of the screen.

To understand the functionality and aesthetics of high-contrast modes, we presented each application twice: (1) distorted images calculated from the reference image, (2) distorted

images calculated from high-contrast screenshots (but the reference left unaltered). The high-contrast images were captured manually, although in the future this could also be automated. We used the *Windows Surface High-Contrast #1 and #2* themes for protanomaly and deuteranomaly simulation, respectively. The only exception was Slack, which has native themes for CVD. In total, there were 120 stimuli (20 UIs × 3 screenshots × 2 modes of standard or high-contrast mode). The trial order was randomized for each participant.

### 3.2. Setup

The experiment was presented on a website. Participants used their own devices, which introduces some variability in terms of screen size, viewing conditions, and color gamut. We postulate that the screen size and viewing condition differences are representative of the real world; however, gamut differences could increase measurement noise.

### 3.3. Task

Once presented with the reference and the three distorted screenshots, participants were asked to rate the following (quotes from the briefing form):

- Functionality on a scale of 1–5: "could you use the distorted user interface the same way as the original reference"?
- Aesthetics as a binary lost/kept: "do the distorted images maintain the aesthetics of the original reference? Aesthetics in this context will be defined as maintaining the same **clarity** and **orderliness** as the reference" (c.f. classical aesthetics). Our preliminary results indicated that participants found the binary scale easier in this instance.

For every new stimulus, both response sliders were initialized to a random location. Participants had unlimited time to adjust the sliders before pressing the submit button.

### 3.4. Participants

In total, 19 people took part in the experiment (age 17–50, 12 male and 7 female), all non-CVD (verified with Ishihara test [39]). Our sample population consisted of people who live in the United Kingdom, work frequently on computers, are familiar with most UIs tested, and are either highly academic or pursuing studies in a selective school. Participation was voluntary without financial incentive, following informed consent. Participants were encouraged to complete all 120 trials; however, this was not a requirement. In total, 7 participants completed all 120 trials, with an average of 79 trials across all participants. As the order of the trials was randomized, this still resulted in a balanced dataset across applications.

### 3.5. Statistical Analysis

Functionality was rated 1–5, and as a common practice with ratings, we treated these linearly to obtain mean opinion scores (MOS) [40]. Then, we first verified that responses in each target population (e.g., an application) passed the Shapiro–Wilk test ($p > 0.05$) of normality as implemented in [41], before estimating the variance and the standard error according to a normal distribution. When checking for significance, we repeatedly applied Welch's two-sided *t* tests across pairs of populations as implemented in [41].

Aesthetics scores were rated as a binary 'kept' or 'lost'. We model each response as the outcome of a Bernoulli experiment with $p$ describing the probability that the aesthetics have been kept. The set of responses for a target population can be modeled as a binomial experiment. Therefore, we estimate the probability $p$ from the samples $x_1, x_2, ..., x_n$ as

$$s' = \frac{1}{n} \sum_{i}^{n} x_i, \tag{1}$$

where $s'$ is the estimated probability of the aesthetics being kept for a given population, $n$ is the number of responses within the sample, and $x_i$ is an individual sample mapped to 1

or 0 if the participant responded 'kept' or 'lost', respectively. We estimate 95% confidence intervals using the normal approximation and perform significance testing across pairs of populations using Fisher's exact test as implemented in [41].

## 4. Results

In the following sections, we distinguish between the following populations when considering functionality scores and the probability of the aesthetics being maintained:

(a) All responses for an individual application (i.e., all participants, including high-contrast and non-high-contrast mode);
(b) Only non-high-contrast mode responses for an individual application;
(c) Only high-contrast mode responses for an individual application;
(d) Non-high-contrast mode responses for all participants and all applications;
(e) High-contrast mode responses for the population of all participants, all applications.

### 4.1. Functionality

Figure 6 summarizes our results comparing populations (a), (b), and (c), with UIs from highest to lowest MOS according to population (a). All presented populations were distributed normally according to the Shapiro–Wilk test. Asterisks indicate when there is a statistical significance between populations (b) and (c) within a single UI (7 our of 20 with $p \ll 0.05$). There was a statistically significant difference in 40 pairings (see Figure 7).

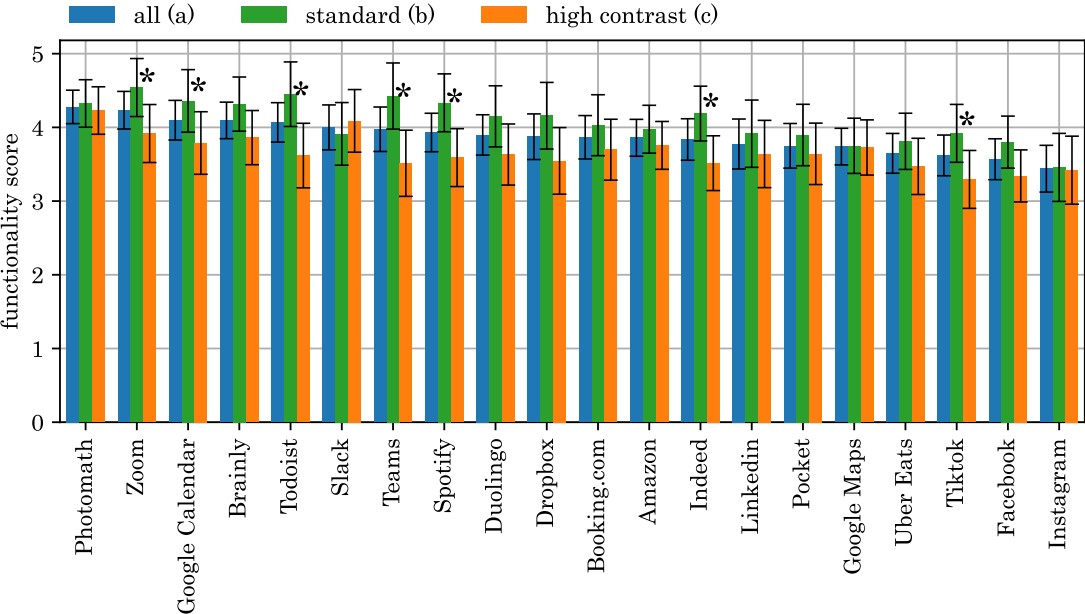

**Figure 6.** Functionality MOS for each UI. Colors show (a) *all* responses, (b) *standard* (non-high-contrast), and (c) *high-contrast* responses for each UI. Error bars indicate 95% confidence intervals, asterisks (*) indicate a significant difference between standard and high-contrast responses for a given app (b vs. c).

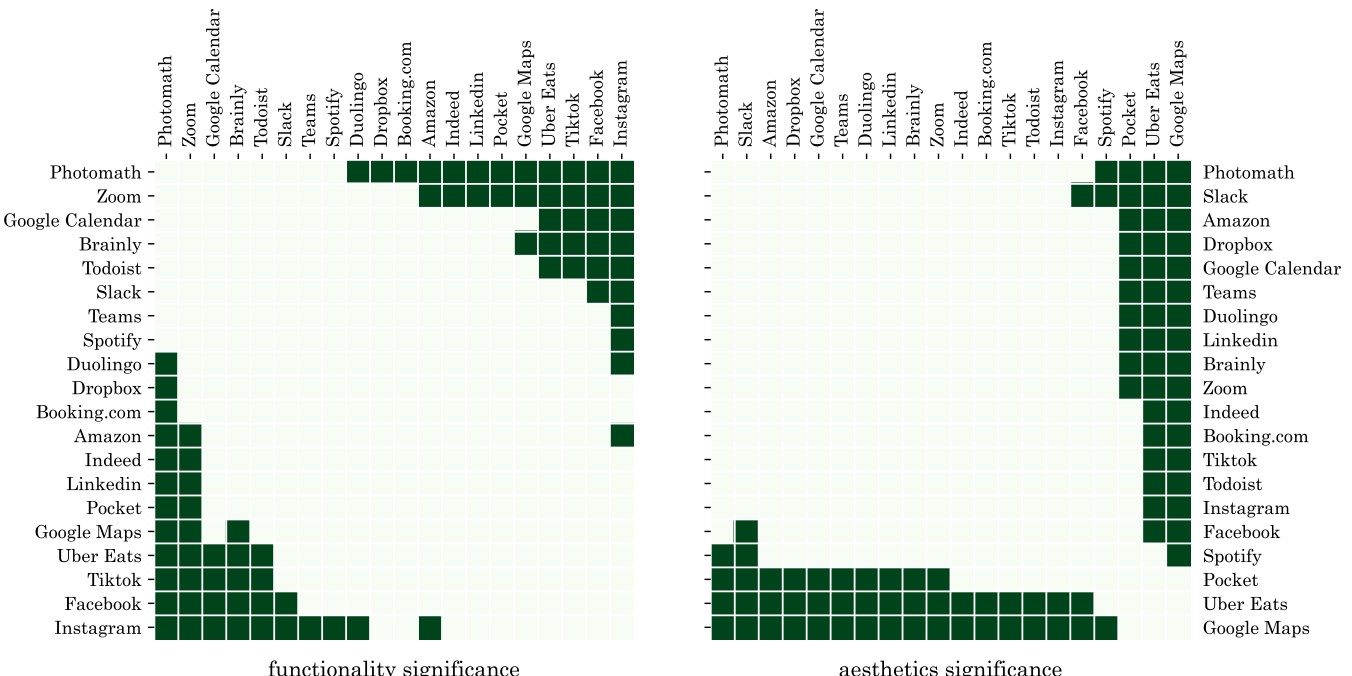

**Figure 7.** Result of significance test on functionality scores and aesthetics with dark squares indicating significance ($p \ll 0.05$). Differences across applications were only significant for highly rated UIs vs. low-rated UIs.

### 4.2. Aesthetics

Figure 8 summarizes the probability of aesthetics kept when comparing populations (a), (b), and (c), with UIs ordered from highest to lowest probability according to population (a). Population (a) probabilities are statistically significant for 46 pairings of applications (see Figure 7). However, what is more noticeable is the marked within-app difference between populations (b) and (c) (standard UI vs. high-contrast stimuli) with the statistical significance test conclusive ($p \ll 0.05$) for 12 out of the 19 UIs tested.

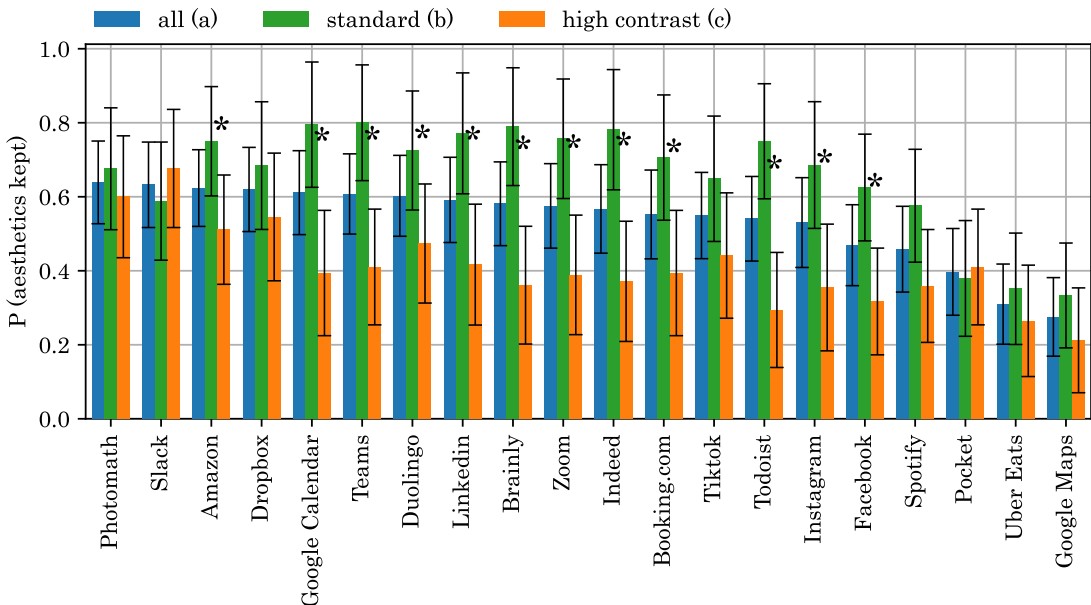

**Figure 8.** Probability of aesthetics being kept for each UI. Colors and notation identical to Figure 6. The difference between (b) and (c) is notable.

*4.3. Functionality vs. Aesthetics*

When investigating the joint population of (b) and (c), i.e., functionality scores and aesthetics probabilities for each app, treating high-contrast and non-high-contrast modes separately, results show a strong correlation (Pearson r = 0.74) between the probability of maintained aesthetics and the functionality score. Figure 9 contains an overview of the results, with more detail available in Figure A1 in Appendix A.

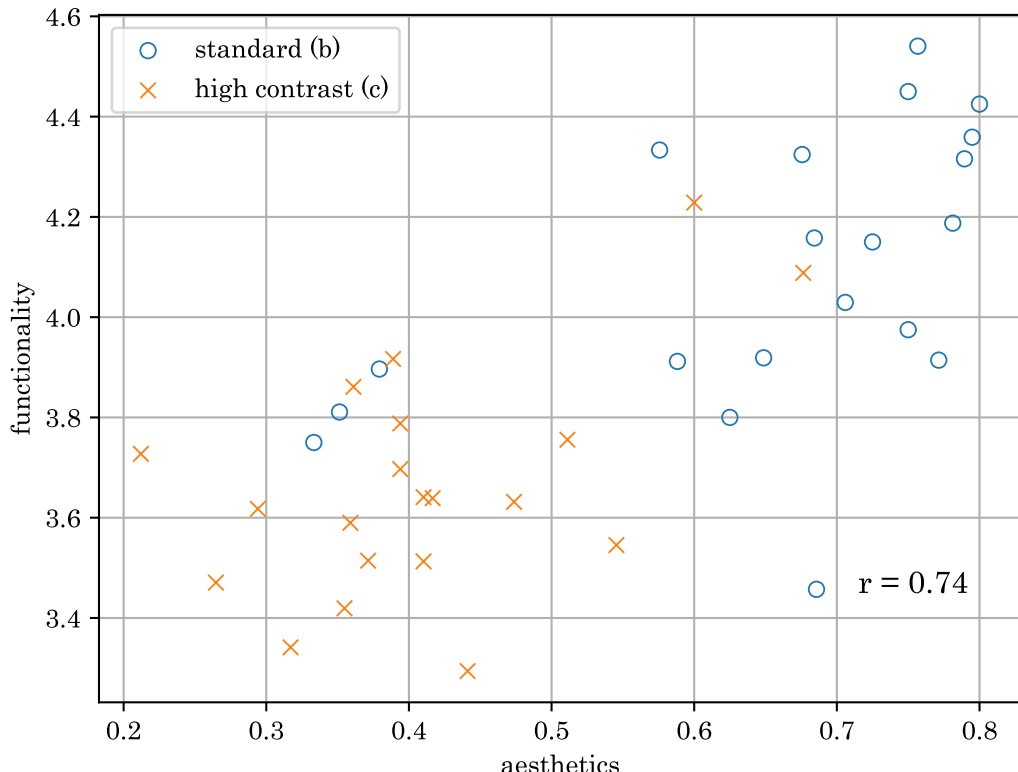

**Figure 9.** Functionality vs. aesthetics for populations (b) *standard* (non-high-contrast) mode (blue circles) and (c) *high constrast* mode (orange crosses). Each point indicates an app. The dataset shows a strong positive correlation (Pearson r = 0.74)

*4.4. High-Contrast Mode*

When comparing populations (d) and (e), i.e., the population of all high-contrast responses vs. the population of all non-high-contrast responses irrespective of applications, functionality scores in both populations pass the Shapiro–Wilk normality test ($p \gg 0.05$), and the functionality scores across the two populations are significantly different according to Welch's two-sided *t*-test ($\mu_{(d)} = 4.09, \mu_{(e)} = 3.66, p \ll 0.05$). Estimated aesthetics probabilities are also significantly different ($s'_{(d)} = 0.66, s'_{(e)} = 0.41$) according to Fisher's exact test ($p \ll 0.05$).

## 5. Discussion

### 5.1. Functionality

Average functionality scores seem favorable (mean = 3.88), especially given the potential bias of non-CVD observers to rate simulated screenshots lower. This could be partially due to the observable positive impact of WCAG; most UIs convey the same information using multiple visual clues (c.f. WCAG 1.4.1), and there is good foreground–background contrast. Analyzing the lower functionality score screenshots reveals a few common pitfalls. Firstly, when color is used to draw users' attention, the impact might be more subtle for a CVD user (see Figure 10). Some UIs seem to use color to differentiate between groups of similar items (e.g., icons), which can make search tasks harder. The final issue we

found seems domain-specific: while many UIs use photos/videos to enrich their design, social media applications rely on these almost exclusively. Such content is naturally more information-rich, and often user-created, which makes it harder to observe accessibility guidelines, e.g., colored text bubbles over a video can be harder to notice by a CVD observer. In the extreme, Instagram received the lowest functionality score in the experiment, which we postulate is because many of Instagram's popular image filters appear mostly identical on the CVD screenshots.

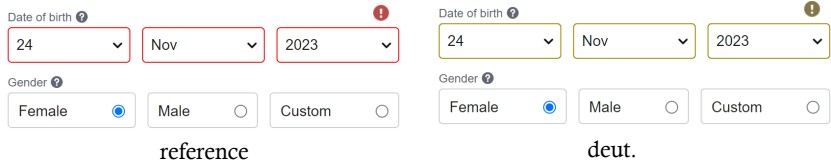

**Figure 10.** Color-coded error field. The UI meets WCAG guidelines with the use of the warning icon; however, the shade of red when seen through CVD simulation (deuteranomaly) stands out less, which can make the user interaction slower in a large form.

*5.2. High-Contrast Mode*

As described in Section 4.4, there is a statistically significant difference in both the functionality scores and the aesthetics preferences when comparing the population of non-high-contrast screenshots with high-contrast screenshots. Figure 9 further reinforces this finding, with high-contrast screenshots rated collectively lower. At least in the context of CVD simulation, we conclude that the Windows high-contrast mode is not suitable for maintaining the aesthetics or the functionality of a UI. Aesthetics is expected, as the high-contrast mode alters the UI with, for example, borders and color contrast that are meant to aid accessibility purely. However, functionality is surprising, as one would expect such accessibility tools to increase functionality rather than decrease it. We speculate that there could be two reasons for the low functionality scores: (1) many interfaces adapt to high-contrast themes poorly, and key visual information can be lost (e.g., see Figure 11), and (2) if there is a linear relationship between aesthetics and functionality, the poor aesthetics scores can have a negative cognitive impact on the perceived functionality as well.

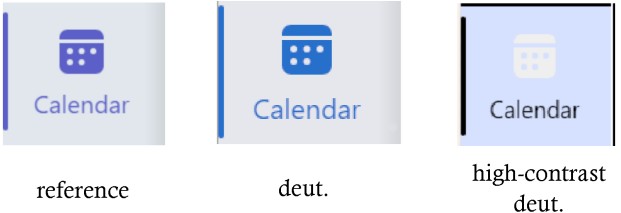

**Figure 11.** Reduced functionality in high-contrast mode. In this instance, turning the calendar icon black would have helped to maintain functionality.

In our sample, the only application where enabling a high-contrast theme had a noticeably positive impact on aesthetics was Slack. However, Slack is also the only application we tested with its own built-in CVD themes (rather than Windows high contrast). This might imply that applications with their own CVD mode could outperform their competitors in terms of aesthetics (when used by someone with CVD). Future research should verify this with a more balanced design of UIs with and without their custom CVD themes.

*5.3. Aesthetics and Functionality*

Results show a strong linear correlation between functionality and aesthetics for the CVD-simulated screenshots when separating the dataset into (b) standard and (c) high contrast. While our scores are relative, they seem to agree with the hypothesis that there

could be a linear relationship between classical aesthetics and functionality. As such, a UI that is tested for CVD functionality can be expected to maintain aesthetics as well.

The only strong outlier in the functionality vs. aesthetics plot was Instagram, with a reasonably high aesthetics score and a surprisingly low functionality score. As discussed, this could be attributed to the inherently challenging task of maintaining functionality in an application, where applying recoloring filters is a core feature.

### 5.4. Classification Framework

Having measured the loss of functionality and aesthetics in a subjective study for a number of UIs, and inspected the relevant screenshots, we propose a simple 3-level classification model to categorize UIs:

- **AAA** functionality and aesthetics are both rated high; functionality scores > 4 and probability of maintained aesthetics > 0.75.
- **AA** aesthetics and functionality are acceptable (functionality > 3.8, aesthetics > 0.5);
- **A** aesthetics might be lost, and functionality is reduced, but the core functionality is still accessible (e.g., through visual means other than color). It meets the WCAG AA accessibility guidelines.

Our classification of the UIs from the experiment is shown in Figure 12.

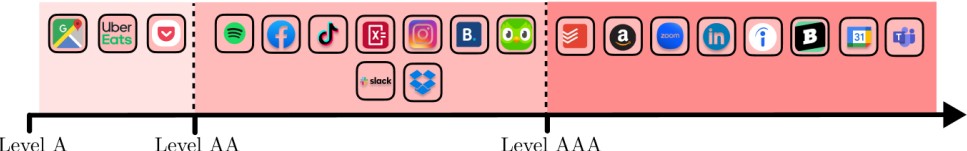

Level A  Level AA  Level AAA

**Figure 12.** Tabular results of our user-interface classification. Software ranked from Level A (lowest) to AAA (highest), where the ordering considers both aesthetics and functionality. Tiles on the same level represent a similar overall score.

### 6. Conclusions and Limitations

CVD affects over 300 million people, but our understanding of how it impacts user-interface (UI) interaction is still limited. In this paper, we discuss the challenges of designing a comparative study, and present a simulation-based protocol measuring if a CVD observer would lose out on any of the functionality or the aesthetics of 20 popular applications/websites as judged by non-CVD observers while inspecting CVD-simulated screenshots. The stimuli dataset and the experiment responses are made available. Our results indicate that UIs that rely on color to distinguish icons or indicate errors might be harder to use for CVD users. Furthermore, we found a linear relationship between functionality and aesthetics scores, which can simplify UI designers' tasks when planning for CVD support. Unfortunately, we also found Windows high-contrast mode to be detrimental to functionality and aesthetics at least in the context of the CVD-simulated screenshots. Results indicate that applications might benefit from implementing custom CVD themes instead, but the significance of this should be tested in future studies.

The biggest limitations of our work are the sample set and bias. Specifically, many popular UIs we analyzed follow similar designs. Evaluating accessibility and aesthetics in a different field (e.g., video games) would require a new set of experiments. Our use of screenshots means that the results are only representative of a subsection of the application. Future studies could consider an interactive setup. Our sample was based in the UK, and aesthetics scores are known to be culturally dependent. Finally, our simulation-based approach can be a powerful way to estimate how a CVD user *might* perceive a UI. However, (1) simulations have known limitations, such as taking place in trichromatic space rather than spectral space, which could alter the accuracy of the simulation, and (2) when judging CVD-simulated images, observers could be biased to object to artifacts

such as discolorations.In future work, we hope to quantify the accuracy of such simulation approaches.

**Author Contributions:** Conceptualization, A.J. and G.D.; methodology, A.J and G.D.; validation, A.J. and G.D.; formal analysis, G.D.; investigation, A.J.; data curation, A.J.; writing—original draft preparation, A.J.; writing—review and editing, A.J. and G.D.; visualization, A.J. and G.D.; supervision, G.D. All authors have read and agreed to the published version of the manuscript.

**Funding:** This research received no external funding.

**Institutional Review Board Statement:** The study was conducted in accordance with the Declaration of Helsinki, and approved by the Head of Research of The Perse School Cambridge (18 November 2023).

**Informed Consent Statement:** Informed consent was obtained from all subjects involved in the study.

**Data Availability Statement:** The data presented in this study are openly available on github: https://github.com/gdenes355/cvd_ui_dataset (accessed on 14 February 2024).

**Acknowledgments:** The authors would like to thank the anonymous reviewers for their detailed, constructive suggestions, and Hazel Knight for her support of this project.

**Conflicts of Interest:** The authors declare no conflicts of interest.

## Appendix A

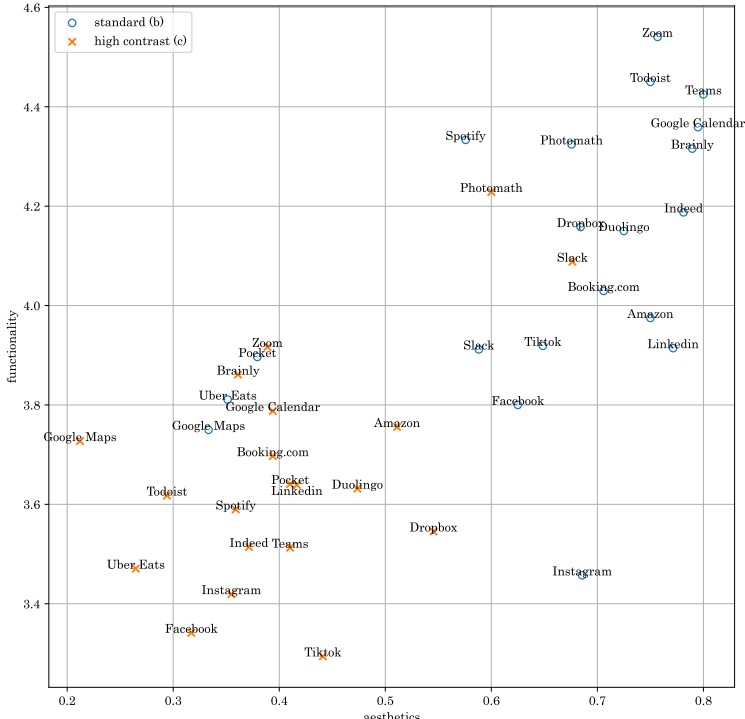

**Figure A1.** Relative functionality scores of simulated CVD screenshots plotted over average probability of aesthetics kept. Each point indicates an app in either (b) *standard* (non-high-contrast) mode or (c) *high-contrast* mode (orange crosses).

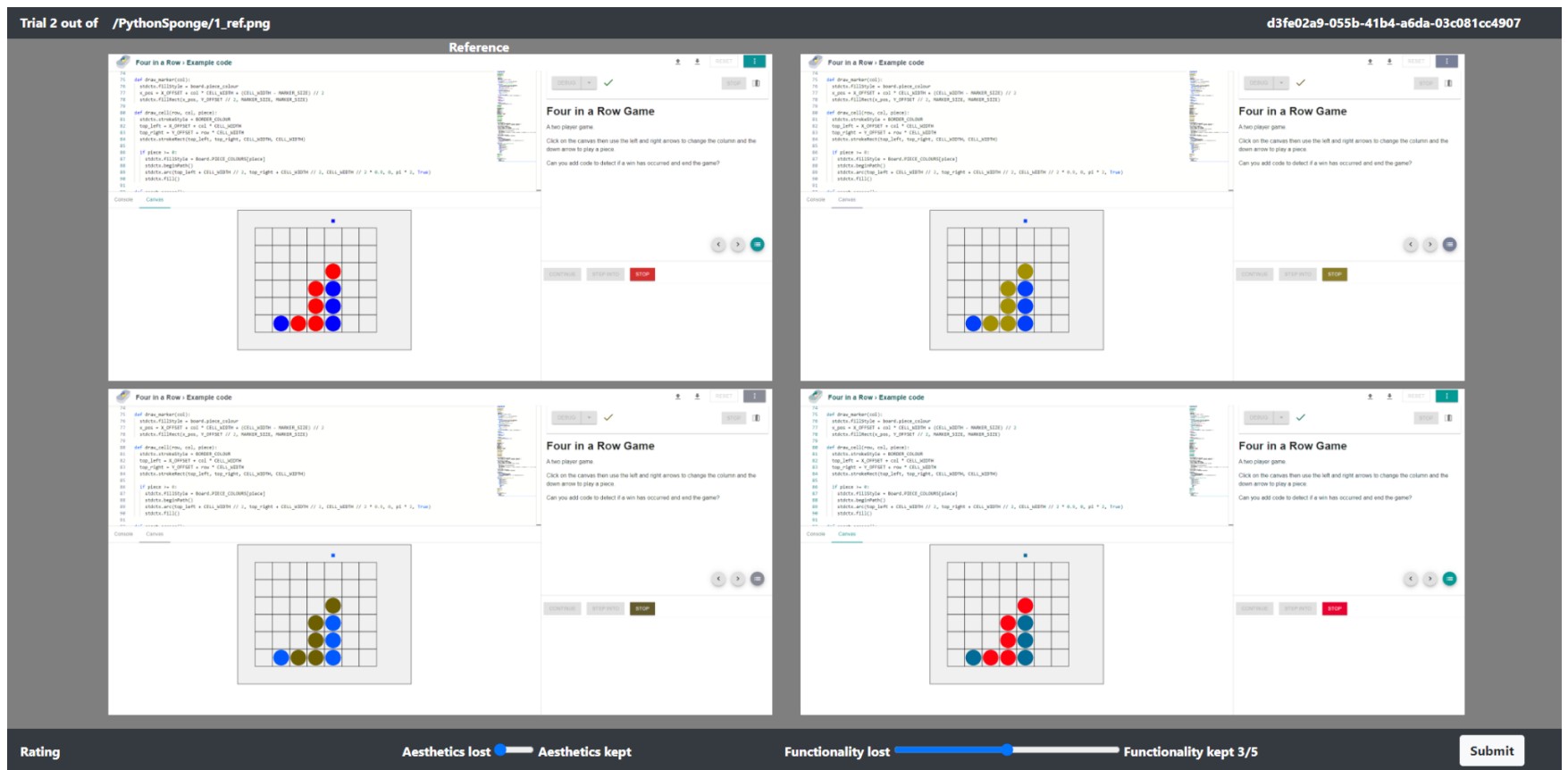

**Figure A2.** Experiment setup: top-left image is the reference screenshot, with three types of CVD (protanomaly: top right, deuteranomaly: bottom left, tritanomaly: bottom right) simulated. The user controls the aesthetics and functionality scores at the bottom of the screen. Hovering over a screenshot magnifies it to reveal it in its original resolution.

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
