# Peer review of "Investigating Color-Blind User-Interface Accessibility via Simulated Interfacesâ€"

_computers, doi:10.3390/computers13020053_

Round 1
Reviewer 1 Report
Comments and Suggestions for Authors
It is an interesting work, however I recommend to organize the presentation of it.
Summary.
The problem of not knowing if user interfaces designed to help people with color blindness meet their objective of allowing a functional interaction with the user is written, and it is also mentioned that this function does not affect the aesthetics.
I believe that the initial wording of the abstract should not be a direct transcription of the introduction section.
However, I recommend describing the method or methodology implemented by the authors to determine the functionality and aesthetics. What population was worked with.
Introduction.
In line 32 it mentions that designers adopt different approaches to address this problem....
The information could be expanded, under what criteria they perform these actions, what studies have been carried out on this selection.
In line 44 it mentions the population of 19 people, this work group has some condition of color blindness, there is a percentage.
Materials
It is recommended to use a diagram or blocks to indicate how to perform the work methodology.
Figure 4, it is recommended that the data should be shown in a clear way for the reader, since showing the capture of four files with data is not appreciated in a correct way.
Results
It is recommended that the data in Figures 5 and 6 be more clearly shown.
It is a little overwhelming to appear with so much data, the results could be reorganized so that the reader has greater ease of understanding the work of the authors.
Author Response
Dear Reviewer,
Thank you very much for taking the time to review this manuscript. Please find the detailed responses below and the corresponding revisions/corrections highlighted in red in the re-submitted files.
Yours faithfully,
The Authors
Summary
I believe that the initial wording of the abstract should not be a direct transcription of the introduction section.
We have updated the abstract according to this recommendation.
However, I recommend describing the method or methodology implemented by the authors to determine the functionality and aesthetics. What population was worked with.
We have now extended our description of the experiment in Section 3, including a more detailed description of the sample population, as well as an inclusion of the statistical methodology.
Introduction
In line 32 it mentions that designers adopt different approaches to address this problem....The information could be expanded, under what criteria they perform these actions, what studies have been carried out on this selection.
Existing approaches can be classified as pre-production color palette reduction or post-production daltonisation, with OS-based high contrast mode being a low-effort mid-ground. We have added appropriate references to the following papers:
https://doi.org/10.1109/ICACS.2018.8333488
https://doi.org/10.1145/1168987.1168996
In line 44 it mentions the population of 19 people, this work group has some condition of color blindness, there is a percentage.
As our experimental procedure relied on a simulation approach, we recruited participants who are not CVD. We verified that participants do not have common forms of CVD using the Ishihara test, which is now highlighted more clearly in the manuscript. We revised the title, the introduction, and the methodology sections of the paper to explain the significance of the simulation-based approach more clearly in the text.
Materials
It is recommended to use a diagram or blocks to indicate how to perform the work methodology.
We have now added a block diagram to summarise the methodology before discussing each step in detail.
Figure 4, it is recommended that the data should be shown in a clear way for the reader, since showing the capture of four files with data is not appreciated in a correct way.
Figure 4 was a screenshot of the original experiment, which we have now moved as a full-page image to appendices as A.2. This gives readers a chance to inspect it in more detail. At the same time, we replaced the original Figure 4 in the main body of the text (now Figure 5) with a schematic diagram of the experiment that the reader will find less cluttered and more informative to understand the experimental setup.
Results
It is recommended that the data in Figures 5 and 6 be more clearly shown. It is a little overwhelming to appear with so much data, the results could be reorganized so that the reader has greater ease of understanding the work of the authors.
Acknowledging, that Figures 5 and 6 were not sufficiently introduced and motivated, we have revised Section 4 to add a clearer introduction to the populations that we visualise and compare. Furthermore, we increased the figure size to aid readability and added Figure 6 that helps the reader to see the results of the repeated t-tests across applications at a glance.
Reviewer 2 Report
Comments and Suggestions for Authors
The paper on CVD in GUIs is an interesting topic that I haven't seen much before.
I didn't find any suggestions for improvement.
I think readers would be interested because this could be relevant to design of any application GUI.
The paper outlines that CVD affects 1 in 12 men so CVD is quite a widespread issue.
I had a question about whether the order of the apps was randomized when presented to the uses, but that was already addressed in the later text.
A good number of common apps have been included in the tests.
I think all spelling and grammar looks good.
A good number of high quality references are all cited correctly and in order.
This seems to be ready to publish in current form.
Author Response
Dear Reviewer,
Thank you very much for taking the time to review this manuscript. We appreciate your positive feedback on the topic as well as the general methodology.
Regarding "I had a question about whether the order of the apps was randomized when presented to the uses, but that was already addressed in the later text.", to aid readability, the manuscript now also clarifies that the stimuli were randomized for each participant at the end of the stimuli section (earlier than previously).
Please find all other revisions/corrections highlighted in red in the re-submitted files.
Yours faithfully,
The Authors
Reviewer 3 Report
Comments and Suggestions for Authors
This work conducted a subjective user study to evaluate CVD-aware website designs in terms of functionality and aesthetics. Experimental stimuli were generated based on CVD simulation and evaluated by participants with normal vision. The authors found that modified stimuli were rated as worse than the reference original stimuli both on functionality and aesthetics dimensions and that the dimensions were positively correlated. In addition, the authors proposed the classification model to categorize UIs.
This paper addresses important issues that support people with CVD in practical situations. The background is well explained. It is plausible that the materials are publicly available. However, the paper needs to justify how subjective ratings by people with normal vision can simulate those of people with CVD. The reviewer is not convinced that the obtained results are generalizable to people with CVD. Therefore, the reviewer does not consider the paper ready for publication at this time.
My major concerns are following:
The reviewer agrees with the difficulties in recruiting a sufficient number of patients with CVD. However, the simulation of the perception of patients with CVD is the focus of the research. In particular, aesthetics is highly dependent on previous life experience. For those who have had normal vision all their lives, it is not surprising that the aesthetics of a reduced color image is less aesthetic. But for those who have had CVD, it may not be the same. It would be best if the authors would consult with ophthalmologists for recruitment and conduct a new experiment with patients. The reviewer is not convinced that their experiment is a simulation of the aesthetic ratings of patients with CVD.
The reviewer could not find the details of the statistics in the paper. The authors compared average functional scores in Figure 5. It was unclear how the authors performed the statistical tests (multiple t-tests or ANOVA?).
The aesthetic scores in Figure 6 range from 0.0 to 1.0. The reviewer could not find the calculation method. Because aesthetics was observed in binary, the score could be interpreted as the probability of choosing ‘lost’ or ‘kept’. If so, the binomial tests could be used in the analysis. However, the test methods were not presented.
The reviewer could not find on what observations the descriptions in sections 4.4 (p. 8) and 5.2 (p. 9) are based.
Author Response
Dear Reviewer,
Thank you very much for taking the time to review this manuscript, and for your constructive, insightful recommendations. Please find the detailed responses below and the corresponding revisions/corrections highlighted in red in the re-submitted files.
Yours faithfully,
The Authors
Summary comments
This paper addresses important issues that support people with CVD in practical situations. The background is well explained. It is plausible that the materials are publicly available. However, the paper needs to justify how subjective ratings by people with normal vision can simulate those of people with CVD. The reviewer is not convinced that the obtained results are generalizable to people with CVD. Therefore, the reviewer does not consider the paper ready for publication at this time.
Regarding the experimental protocol and the use of CVD simulation instead of recruiting CVD participants: we believe that a comparative study would be non-trivial, as CVD observers cannot judge what they cannot see any more than a non-CVD participant cannot judge what they might not see (without simulation). Even in a large-scale study with subjective ratings from all groups of CVD and non-CVD as a control group, one could argue that participants’ experiences would be strongly biased by current UI design, and judgement of aesthetics might be biased differently across the groups. We look forward to investigating the matter further; however, for the scope of this article, we propose keeping the simulation-based methodology and hence we rewrote the manuscript carefully to signpost related limitations, re-worded claims and conclusions to avoid misleading the reader. As part of the process, we have now updated the title to “Investigating Color Blind User Interface Accessibility via Simulated Interfaces”
Major concerns
The reviewer agrees with the difficulties in recruiting a sufficient number of patients with CVD. However, the simulation of the perception of patients with CVD is the focus of the research. In particular, aesthetics is highly dependent on previous life experience. For those who have had normal vision all their lives, it is not surprising that the aesthetics of a reduced color image is less aesthetic. But for those who have had CVD, it may not be the same. It would be best if the authors would consult with ophthalmologists for recruitment and conduct a new experiment with patients. The reviewer is not convinced that their experiment is a simulation of the aesthetic ratings of patients with CVD.
We agree that the simulation-based approach is not a perfect solution, and therefore results should not be automatically generalisable to the population of people with CVD. However, we also argue that there are numerous other challenges in a comparative experimental methodology beyond participant recruitment. First of all, CVD participants could not make comparative judgements, so a hypothetical experiment would need to based on subjective ratings of individual UIs. As aesthetics is indeed highly dependent on previous life experiences, we do not see any guarantees that for instance aesthetic scores from the population of people with deuteranopia could be directly compared to the aesthetic scores of someone without CVD. For functionality, a task performance based comparison might be feasible, and we look forward to investigating this in future work.
The main limitation of the simulation-based approach is indeed that ratings are relative (as pointed out by the reviewer, it would be unrealistic to expect non-CVD observers to not anchor the full-color UIs above even the best CVD-simulated images). However, we argue that relative ratings are still meaningful in the sense that the scores allowed us to investigate how different applications and strategies might succeed or fail to maintain functionality and aesthetics at least in the context of CVD simulations. For example, our findings for the loss of functionality for high contrast mode (vs. non-high contrast mode) are surprising, and seem relevant irrespective of the experimental methodology. We propose that such contributions are valuable to the community, they can inspire future work and can help the industry to reflect on current practices while the next experimental methodology is developed.
We also value academic integrity, and we have now majorly revised the manuscript to highlight the limitation of the simulation-based approach more clearly. We also updated the title, the abstract, the introduction, the analysis and the summary to avoid misleading the reader and to better explain the scope of our work.
The reviewer could not find the details of the statistics in the paper. The authors compared average functional scores in Figure 5. It was unclear how the authors performed the statistical tests (multiple t-tests or ANOVA?).
We performed multiple t-tests to compare significance across different populations (e.g. across pairs off apps, or across high contrast vs non-high contrast). We acknowledge that this should have been explained with more clarity and added a new Section 3.5 in the methodology section to describe the details of the statistical models applied.
The aesthetic scores in Figure 6 range from 0.0 to 1.0. The reviewer could not find the calculation method. Because aesthetics was observed in binary, the score could be interpreted as the probability of choosing ‘lost’ or ‘kept’. If so, the binomial tests could be used in the analysis. However, the test methods were not presented.
The statistical tests used for the Binomial aesthetic scores are also described now in Section 3.5.
The reviewer could not find on what observations the descriptions in sections 4.4 (p. 8) and 5.2 (p. 9) are based.
The claims in Section 4.4 are based on the t-test between the population of high-contrast vs. Non-high-contrast responses for functionality scores, and Fisher’s exact test for the aesthetics scores. Intuitively, Figure 7 also shows a remarkable difference between high-contrast and non-high-contrast responses.
The descriptions in Section 5.2 follow from 4.4. We believe that the claims in Section 4.4 are now more clearly reinforced by experimental data, which should also add to the quality of the description in Section 5.2. We have revised Section 5.2 to make this link clearer.
Round 2
Reviewer 3 Report
Comments and Suggestions for Authors
Thank you for the revision.
The reviewer has determined that sufficient justification for the study and technical soundness are provided.
I believe that this manuscript is ready for publication.